# Hypertriglyceridemia Induced Acute Pancreatitis Caused by a Novel LIPC Gene Variant in a Pediatric Patient

**DOI:** 10.3390/children9020188

**Published:** 2022-02-02

**Authors:** Laura Balanescu, Ancuta Cardoneanu, Gabriel Stanciu, Radu Balanescu, Cristian Minulescu, Daniela Pacurar, Andreea Moga

**Affiliations:** 1Pediatric Surgery Department, Grigore Alexandrescu Emergency Hospital for Children, 011743 Bucharest, Romania; laura7balanescu@yahoo.com (L.B.); stanciu.gabriel_daniel@yahoo.com (G.S.); radu.balanescu@umfcd.ro (R.B.); danapacurar@yahoo.com (D.P.); andreea.moga@ymail.com (A.M.); 2Department of Pediatric Surgery and Orthopedics, Carol Davila University of Medicine and Pharmacy, 050474 Bucharest, Romania; minulescucristian@yahoo.com

**Keywords:** LIPC gene variant, hypertriglyceridemia, pediatric acute pancreatitis, dyslipidemia

## Abstract

Hypertriglyceridemia induced acute pancreatitis is a rare cause of pancreatitis in children. Hepatic lipase deficiency is an extremely rare cause of hypertriglyceridemia, reported in only a few families to date. Hepatic lipase is the enzyme involved in the hydrolysis of triglycerides and phospholipids in remnants of triglyceride-rich lipoproteins that have a role in the conversion of very low density lipoprotein remnants to low density lipoproteins. Hepatic lipase deficiency is inherited in an autosomal recessive pattern. Detection of heterozygous carriers of hepatic lipase mutations remains accidental at the population level, as affected persons with a heterozygous state of hepatic lipase mutation do not display specific lipoprotein abnormalities and also patients with complete hepatic lipase deficiency have inconstant phenotype. The proximal promoter of the LIPC gene consists of four polymorphic sites in complete linkage disequilibrium. Five missense mutations in encoding exons have been described and proved to be responsible for hepatic lipase deficiency to date: S267F, T383M, L334F, A174T, and R186H, affecting the activity and secretion of hepatic lipase. We identified a primary disorder of the lipid metabolism as the cause of the acute episode of pancreatitis in a four years old patient, consisting of hepatic lipase deficiency caused by a novel genetic variant of the LIPC gene, a gross deletion of the genomic region encompassing exon 1. This variant was not previously described in the literature in persons with LIPC-related disorders and its significance is currently uncertain, but in the presented clinical and paraclinical context, it has the characteristics of a pathological variant inducing a hepatic lipase deficiency phenotype.

## 1. Introduction

Primary diseases of lipid metabolism causing hypertriglyceridemia derive from genetic anomalies in triglyceride synthesis and metabolism [1]. Hypertriglyceridemia induced acute pancreatitis is an extremely rare, but important cause of pancreatitis in the pediatric population because it leads to significant morbidity and mortality [2]. An extremely rare cause of hypertriglyceridemia is hepatic lipase deficiency, an autosomal recessive transmitted disease that has been reported only in a few families to date [3]. Hepatic lipase is a lipolytic serine hydrolase synthesized and secreted by the hepatic cells, attached to the liver sinusoidal surface by heparin sulphate proteoglycans [4]. Only a few mutations and gene polymorphisms of hepatic lipase gene were reported [5]. We present a hypertriglyceridemia induced acute pancreatitis case in a pediatric patient caused by a previously unreported LIPC gene variant, a gross deletion of the genomic region encompassing exon 1 of the LIPC gene.

The aim of this paper is to present the rare case of acute pancreatitis in a pediatric patient determined by an extremely rare cause of hypertriglyceridemia, a novel variant of the LIPC gene, reported in only a few families to date.

## 2. Case Report

A 4 years and 9 months old boy presented to our Emergency Care Unit after 24 h of abdominal pain, anorexia, nausea, and bilious vomiting at home. Initial monitoring of the patient revealed a temperature of 36.5 C, heart rate 97 bpm, blood pressure 119/83 mmHg, respiratory rate 22 breaths/min, and an oxygen saturation of 98% on room air. Initial blood analysis showed leukocytosis with neutrophilia (22.14 × 109/L and 18.73 × 109/L) and elevated inflammatory markers (ESR 60 mm/1 h, CRP 2.4 mg/dL). Blood gas analysis and biochemical analysis could not be initially achieved due to the high lactescent serum. Lipid profile was highly modified, with total lipids counting 3441 mg/dL with total cholesterol of 923 mg/dL and triglycerides level of 1274 mg/dL. Initial abdominal ultrasound identified a hyperechoic pancreas without any other pathological aspects. Thoraco-abdominal X-ray was normal.

The patient was admitted to the Pediatric Department of our hospital for management and further investigations.

His physical examination identified no xanthelasma or skin eruptions. The patient has a BMI of 20.1 (29 Kg, 118 cm), above 95th percentile according to CDC (Center for Disease Control and Prevention). Abdominal examination revealed a non-distended abdomen with mild tenderness and diffuse sensibility, accentuated in the periumbilical region. The patient was previously healthy without any medical or surgical history. The patient has a defective diet, with more than six meals every day, including fast-food and other high-processed foods and drinks. There was no history of medication, infection, recent surgeries, gastrointestinal endoscopic procedures, or abdominal trauma. There were no similar episodes previously. There was no family history of pancreatitis or gallstones. His mother, 32 years, suffers from migraines. Apart from that, she has no medical history and her lipidic profile is normal. His father, 38 years old, has mixt dyslipidemia. He has an 11-year-old brother without any significant medical history. His maternal grandfather has type 2 diabetes mellitus.

The complete blood analyses obtained after admission in the Pediatric Department also revealed a mild anemia (Hb 11.8 g/dL) and elevated pancreatic enzymes (amylase 76 U/L, lipase 69 U/L), that reached 153 U/l and, respectively 176 U/L within a week from admission. Liver enzymes including ALT, AST, gamma-GT, ALP, and total bilirubin levels were within referenced ranges. Random glucose level was 124 mg/dL. A fasting lipid panel was collected, and the most striking abnormality was a marked elevation of triglycerides (1359 mg/dL) and total cholesterol (538 mg mg/dL) with HDL- 35.6 mg/dL, LDL- 214.8 mg/dL, and VLDL- 54 mg/dL. Blood gas analysis identified a pH level of 7.56, PCO2- 24 mm Hg, PO2- 152 mmHg, and HCO3- 24.7 mmol/L. Serum electrolytes and coagulation tests were within normal ranges. An abdominal computed tomography was performed and was compatible with the diagnosis of acute pancreatitis involving the entire pancreas, with significant fat stranding, but without any systematized fluid collections. The other abdominal organs were within normal limits (Figure 1). Cardiovascular exam was unremarkable.

As the general condition of the patient continued to deteriorate under medical treatment and there was increasing patient distress, he was transferred into the Intensive Care Unit for supportive measures and management. Patient management included non-oral intake, intravenous rehydration, antibiotics, antiemetics, antacids, pain medication, statins, fibrates, and pancreatin supplement. Clinical and biologic status of the patient slowly improved, and he was readmitted to the Pediatric Department. Periodical abdominal ultrasonography identified the evolution of the pancreatic inflammation into a pancreatic pseudocyst supplementary characterized by computed tomography 13 days after admission as a large peripancreatic collection of 88/85/131 mm (T/AP/CC) (Figure 1). Endoscopic and laparoscopic drainage were attempted, but without satisfactory evacuation of the collection, a reintervention for external drainage of the pseudocyst being required with favorable evolution. The drainage was removed on postoperative day 10. Computed tomography reevaluation after 30 days showed the collection measuring 23/33/35 mm (T/AP/CC). Ultrasonographic reevaluation 60 days postoperatively did not identify any collection.

As the acute event was successfully treated, it was mandatory to focus on its cause to prevent recurrent episodes of pancreatitis.

Initial complete lipid profile identified important hypertriglyceridemia (1359 mg/dL) and hypercholesterolemia (538 mg mg/dL) with LDL predominance (214.8 mg/dL). Lipoprotein electrophoresis revealed hyper-β-lipoproteinemia and hypo-α-lipoproteinemia. Triglyceride levels decreased under diet and fibrate treatment and reached 214 mg/dL after 30 days. As these values were too elevated to be only secondary to the acute episode of pancreatitis, a primary dyslipidemia was considered. Sequence analysis and deletion/duplication testing using Next-Generation Sequencing technologies was performed on a panel of 36 genes involved in lipid metabolism (Table 1). DNA was extracted from peripheral blood. A gross deletion of the genomic region encompassing exon 1 of the LIPC gene, which includes the initial codon, was identified. The 5′ end of this event is unknown, as it extends beyond the assayed region for this gene and therefore may encompass additional genes. The 3′ boundary is likely confined to intron 1 of the LIPC gene. It is anticipated to produce a missing or disrupted protein product. However, the available clinical and genetic data is not adequate to demonstrate whether loss-of-function variants in LIPC cause disease. This variant was not previously reported in the literature in persons with LIPC-related disorders, and its significance is currently uncertain, but in the presented clinical and paraclinical context, it has the characteristics of a pathological variant inducing a hepatic lipase deficiency phenotype.

## 3. Discussion

Primary disorders of lipid metabolism inducing hypertriglyceridemia derive from genetic errors in triglyceride production and metabolism. Triglyceride levels above 500 mg/dL are encountered in less than 0.2% of cases, but when detected, an immediate concern of a primary triglyceride metabolism disorder should be considered [1].

The global incidence of acute pancreatitis in pediatric population is ascending, having an estimated incidence of 1/10,000 children annually for pancreatitis determined by all causes [6]. In the pediatric population, etiologies of acute pancreatitis are varied, including genetic disorders, systemic conditions, infectious diseases, autoimmune disorders, metabolic causes, drugs and toxins, obstructive causes, and trauma [7]. The specific incidence of hypertriglyceridemia induced acute pancreatitis is currently unknown due to its rarity, being mostly presented as case reports [2]. Although the precise mechanism of pancreatitis is not currently established, it is thought that triglyceride-rich chylomicrons decrease circulatory stream in capillary beds of the pancreas leading to ischemia and generating an inflammatory response [8]. Pancreatitis usually appears when triglyceride levels are above 1000 to 1500 mg/dL, but triglyceride levels from 200 to 1000 mg/dL can be detected in the initial period of acute pancreatitis of all etiologies [2,9]. In our case, the patient had an initial triglyceride level of 1274 mg/dL.

Once the diagnosis is established and the acute episode treated, prevention of pancreatitis is essential as mortality caused by pancreatitis can reach 11% [10]. Prevention of pancreatitis of this etiology is based on triglyceride lowering below 500 mg/dL. In our case, we managed to obtain a triglyceride level of 214 mg/dL after 30 days of diet and fibrate treatment. Usually, recurrent episodes of pancreatitis appear because the compliance with a low-fat diet is challenging and pharmacologic therapy is not constantly efficient. Additionally, pediatric patients with constant moderate-to-high levels of triglyceride can be at high risk of early cardiovascular disease during maturity [1]. In our patient, the cardiovascular exam was normal, but he will be periodically reevaluated for the occurrence of any signs of cardiovascular disease.

We identified a primary disorder of the lipid metabolism as the cause of the acute episode of pancreatitis in our patient, consisting of a hepatic lipase deficiency caused by a novel genetic variant of the LIPC gene, a gross deletion of the genomic region encompassing exon 1. Hepatic lipase (HL) is an essential enzyme in the metabolism of triglyceride and phospholipids [11]. HL is a lipolytic serine hydrolase encrypt by the LIPC gene, produced and secreted by liver cells and attached to the liver sinusoidal area by heparin sulfate proteoglycans [12]. Establishment of the role of HL in human lipid metabolism has been simplified by the recognition of individuals with HL deficiency. HL is involved in the hydrolysis of triglycerides and phospholipids in remnants of triglyceride-rich lipoproteins that have a role in the transformation of VLDL remnants to LDL [13]. HL also promotes the incorporation of these particles, acting as their ligand in liver cells. Additionally, HL plays a role in the transformation of large buoyant high-density lipoprotein (HDL) to small HDL influencing the phospholipids load of these remnants [13]. HL is discharged from the surface of the hepatocytes into circulation by the intravenous injection of heparin and usually evaluated in humans as enzyme activity in post heparin plasma [14,15].

The LIPC gene is found on chromosome 15q21-q23 lengthening 35 kb and it consists of a total of nine exons and eight introns encoding a 476 amino acid glycoprotein lipolytic serine hydrolase [14,16]. The HL gene, or LIPC gene, is a member of the lipase gene family [17]. The three distinct lipases, lipoprotein lipase (LPL), hepatic lipase (HL), and pancreatic lipase (PL), have a considerable amount of similarities in their protein sequences, especially proximal to the three amino acids that form the catalytic triad involved in enzyme activity [18,19]. LIPC gene is expressed, even if to variable levels, in nearly all species [20]. Human HL has appreciable DNA and protein sequence homology with HL of rat and rabbit [21].

The proximal promoter of the LIPC gene consists of four polymorphic sites in complete linkage disequilibrium: G-250A, C-514T, T-710C, and A-763G, as established by the nomenclature of Amesis et al. There have been reported correlations between variations in plasma biochemical traits and variations in the promoter sequence of the LIPC gene [22]. Several variations of the LIPC gene have been reported (Table 2). Polymorphisms in the promoter sequence of LIPC gene can cause or not cause amino acid substitutions. Polymorphisms causing amino acid substitutions are V73M, N193S, S267F, T383M, L334F, and R186H. Polymorphisms that do not lead to amino acid change were reported in codons V133V, T202T, T457T, G175G, and T344T. Moreover, a C-toT substitution in nucleotide -480 in the promoter region of the LIPC gene and a mutation in intron 1 have been reported [5]. The polymorphism C-514T in the promoter region explains about 38% of the inconsistency of the HL activity [23]. There are also fewer common variants, and their frequency was reported to be between 0.15 to 0.21 among Caucasians, 0.45 to 0.53 among African Americans, and 0.47 among Japanese Americans [23,24].

Five missense mutations in encoding exons were reported and proved to determine HL deficiency to date: S267F, T383M, L334F, A174T, and R186H [3,26,27,28], Al [29].

The practical importance of these mutations was established by site directed mutagenesis and in vitro expression analysis. The S276F mutation seems to critically influence the activity and releasing of HL, while the T383M mutant variant maintains partial activity, but is inadequately secreted [30]. The L334F mutation determines almost normal production of a HL product that has only around 30% of the wild type protein function [27]. The R186H mutation determines arginine to histidine substitution in the mature protein producing an inactive HL protein [27]. The A174T mutation was associated with T383M mutation in three individuals of the Quebec family who had extremely low to undetectable HL activity [4].

The T383M (thr383-to-met) substitution in the mature enzyme caused by a 1221C-T transition in exon 8 of the LIPC gene was described by Hegele et al. in 1991 in six individuals with complete HL deficiency from two unrelated families, one of French descent living in Quebec and a second of Irish and English descent living in Ontario. Hegele et al. in 1991 identified in three affected individuals of the Ontario family another mutation in the LIPC gene, an 873C-T transition in exon 8 resulting in S267F (ser-267-to-phe) substitution. Ruel et al. identified in 2003 a second mutation in affected members of the Quebec family, a G-to-A transition in exon 5 resulting in A174T (ala174-to-thr) substitution in a highly conserved region of the mature product. Knudsen et al. described in 1996, in a Finnish man with HL deficiency, a heterozygous mutation in the LIPC gene: R186H and L334F substitution in exons 5 and 7 of the HL gene. In 2010 Al Riyami et al. identified homozygosity for the L334F (leu334-to-phr) mutation in the LIPC gene in an Arab patient born with consanguineous parents with HL deficiency; post heparin plasma HL activity in this patient was zero [3,26,27,28,29].

This extremely rare genetic disease, which seems to be transmitted in an autosomal recessive manner, was described only in five families to date [31,32,33,34,35]. HL deficiency is transmitted in an autosomal recessive manner; thereby, both copies of the gene in each cell have mutations. Both parents of a person with an autosomal recessive disorder are carriers of one copy of the mutated gene, but they usually do not present signs of the disease [36]. Its real prevalence might be significantly higher due to the struggle in diagnosis by conventional clinical and biochemical criteria [11]. In clinical practice, the usual lipid markers such as serum total LDL and HDL cholesterol and triglycerides do not reveal any specific aberrations. Therefore, detection of heterozygous carriers of HL mutations is incidental at population level [27]. In the case of heterozygous carriers, functional mutations can determine a clinical impact only if there are interactions between environment factors and other genes [27]. Affected persons that have a heterozygous state for HL mutations do not develop characteristic lipoprotein anomalies, and even individuals with complete HL deficiency show an inconstant phenotype [3]. Data from HL deficient patients indicate that secondary factors such as age, gender, and visceral obesity with insulin resistance can modify the phenotypic manifestations of HL mutations [3].

By influencing the phospholipid and triglyceride load of IDL, LDL, and HDL, HL plays and important role in establishing their lipid composition, density, size and by that their metabolic fate [3]. The most frequent consequences of HL deficiency on lipoprotein profiles have been triglyceride enrichment of LDL and HDL, presence of beta-VLDL and modified catabolism of chylomicrons [37]. This phenotype is similar to type III hyperlipoproteinemia [11]. Our patient presented with hypertriglyceridemia and hypercholesterolemia with LDL predominance. The limitations of our study include that we present a singular case and our report may therefore be taken with reservations when extrapolated to other patients and long term consequences of this variant in our patient are not available to report at the moment and can only be predicted based on other studies.

## 4. Conclusions

We report a pediatric case of acute pancreatitis caused by a previously unknown variant of the LIPC gene, a gross deletion of the genomic region encompassing exon 1 of the LIPC gene resulting in hypertriglyceridemia and hypercholesterolemia.

## Figures and Tables

**Figure 1 children-09-00188-f001:**
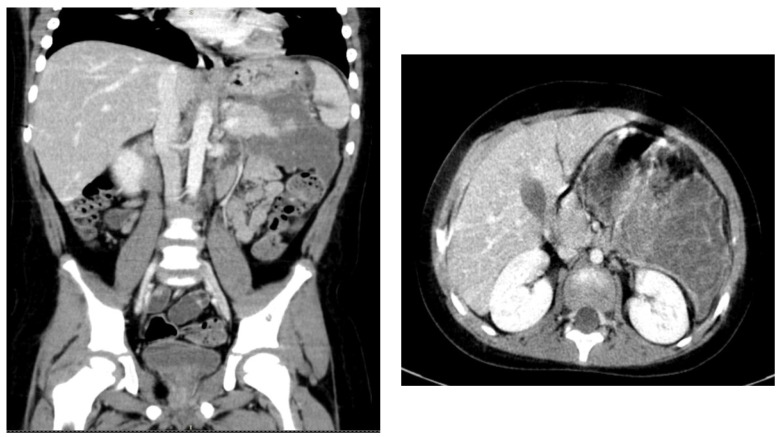
Computed tomography aspect- pancreatic pseudocyst.

**Table 1 children-09-00188-t001:** List of lipid metabolism involved genes analyzed for our patient.

ABCA1	LCAT
ABCG5	LDLR
ABCG8	LDLRAP1
ANGPTL3	LIPA
APOA1	LIPC
APOA4	LIPG
APOA5	LIPI
APOB	LMF1
APOC2	LPL
APOC3	LRP6
CETP	MTTP
CREB3L3	MYLIP
CYP27A1	PCSK9
CYP7A1	PLTP
GALNT2	PNPLA2
GCKR	SAR1B
GPD1	SCARB1
GIPHBP1	ZHX3

**Table 2 children-09-00188-t002:** LIPC gene variants [25].

Nr.	Name	Significance
1	NM_000236.3 (LIPC): c.583G > A (p.Ala195Thr)	Pathogenic
2	L334F	Pathogenic
3	NM_000236.3 (LIPC): c.1214C > T (p.Thr405Met)	Uncertain significance
4	NM_000236.3 (LIPC): c.1226A > C (p.Asp409Ala)	Uncertain significance
5	NM_000236.3 (LIPC): c.787A > G (p.Ile263Val)	Uncertain significance
6	NM_000236.3 (LIPC): c.738_739dup (p.Gly247fs)	Uncertain significance
7	NM_000236.3 (LIPC): c.866C > T (p.Ser289Phe)	Uncertain significance
8	NM_000236.3 (LIPC): c.986G > A (p.Arg329His)	Uncertain significance
9	NM_000236.3 (LIPC): c.67C > T (p.Leu23Phe)	Uncertain significance
10	NM_000236.3 (LIPC): c.88 + 5G > C	Uncertain significance
11	NM_000236.3 (LIPC): c.132G > A (p.Thr44=)	Uncertain significance
12	NM_000236.3 (LIPC): c.207G > A (p.Pro69=)	Uncertain significance
13	NM_000236.3 (LIPC): c.314C > T (p.Ala105Val)	Uncertain significance
14	NM_000236.3 (LIPC): c.403C > T (p.Arg135Cys)	Uncertain significance
15	NM_000236.3 (LIPC): c.456 + 5C > T	Uncertain significance
16	NM_000236.3 (LIPC): c.1170G > A (p.Leu390=)	Uncertain significance
17	NM_000236.3 (LIPC): c.1215G > T (p.Thr405=)	Uncertain significance
18	NM_000236.3 (LIPC): c.1232G > T (p.Gly411Val)	Uncertain significance
19	NM_000236.3 (LIPC): c.867C > T (p.Ser289=)	Uncertain significance
20	NM_000236.3 (LIPC): c.1052-13TC [6]	Uncertain significance
21	NM_000236.3 (LIPC): c.1231G > A (p.Gly411Ser)	Uncertain significance
22	NM_000236.3 (LIPC): c.316G > A (p.Ala106Thr)	Uncertain significance
23	NM_000236.3 (LIPC): c.206C > T (p.Pro69Leu)	Uncertain significance
24	NM_000236.3 (LIPC): c.1430G > A (p.Arg477His)	Uncertain significance
25	NM_000236.3 (LIPC): c.588G > A (p.Ala196=)	Uncertain significance
26	NM_000236.3 (LIPC): c.317C > T (p.Ala106Val)	Uncertain significance
27	NM_000236.3 (LIPC): c.1169 + 11G > A	Uncertain significance
28	NM_000236.3 (LIPC): c.998G > A (p.Arg333Gln)	Uncertain significance
29	NM_000236.3 (LIPC): c.739G > A (p.Gly247Arg)	Uncertain significance
30	NM_000236.3 (LIPC): c.1341C > T (p.Gly447=)	Uncertain significance
31	NM_000236.3 (LIPC): c.461C > A (p.Ser154Tyr)	Uncertain significance
32	NM_000236.3 (LIPC): c.575-5A > G	Uncertain significance
33	NM_000236.3 (LIPC): c.1203C > T (p.Ser401=)	Uncertain significance
34	NM_000236.3 (LIPC): c.1388 + 13T > G	Uncertain significance
35	NM_000236.3 (LIPC): c.1415A > T (p.Asp472Val)	Uncertain significance
36	NM_000236.3 (LIPC): c.1421T > C (p.Leu474Pro)	Uncertain significance
37	NM_000236.3 (LIPC): c.829T > C (p.Cys277Arg)	Uncertain significance
38	NM_000236.3 (LIPC): c.1273G > C (p.Val425Leu)	Uncertain significance
39	NM_000236.3 (LIPC): c.947G > T (p.Cys316Phe)	Uncertain significance
40	NM_000236.3 (LIPC): c.981C > T (p.His327=)	Likely benign
41	NM_000236.3 (LIPC): c.1314A > G (p.Pro438=)	Likely benign
42	NM_000236.3 (LIPC): c.575-15C > T	Likely benign
43	NM_000236.3 (LIPC): c.1233C > T (p.Gly411=)	Likely benign
44	NM_000236.3 (LIPC): c.1052-10C > G	Likely benign
45	NM_000236.3 (LIPC): c.213G > A (p.Thr71=)	Likely benign
46	NM_000236.3 (LIPC): c.1029G > A (p.Thr343=)	Likely benign
47	NM_000236.3 (LIPC): c.1064A > G (p.Gln355Arg)	Benign
48	NM_000236.3 (LIPC): c.1098A > G (p.Thr366=)	Benign
49	NM_000236.3 (LIPC): c.644A > G (p.Asn215Ser)	Benign
50	NM_000236.3 (LIPC): c.264C > T (p.His88=)	Benign
51	NM_000236.3 (LIPC): c.591A > G (p.Gly197=)	Benign
52	NM_000236.3 (LIPC): c.1437C > A (p.Thr479=)	Benign
53	NM_000236.3 (LIPC): c.672C > G (p.Thr224=)	Benign
54	NM_000236.3 (LIPC): c.465G > T (p.Val155=)	Benign
55	NM_000236.3 (LIPC): c.575-8C > A	Benign
56	NM_000236.3 (LIPC): c.1068C > A (p.Phe356Leu)	Benign
57	NM_000236.3 (LIPC): c.283G > A (p.Val95Met)	Benign
58	NM_000236.3 (LIPC): c.837C > T (p.His279=)	Benign

## Data Availability

Data is available at cardoneanu.anca@gmail.com if needed.

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
