# Peer review of "Hypertriglyceridemia Induced Acute Pancreatitis Caused by a Novel LIPC Gene Variant in a Pediatric Patient"

_children, 2022, doi:10.3390/children9020188_

Round 1

Reviewer 1 Report

Authors have described the case of AP from disorders of lipid metabolism. 

Please mention where he had abdominal pain? epigastric or any other region. Any radiating pain to the back? 

Was fasting lipid panel collected eventually?

What are the normal values for amylase and lipase in your lab? Please include those. Usually 2 to 3 times the upper limit of normal is considered significant for a diagnosis of pancreatitis. These values are less likely to be acute pancreatitis as even persistent emesis can give rise to these elevated values. 

Please mention if he qualifies for Morbid or class of obesity? Class I or 2 or 3?

what is diffuse sensibility? I apologize as I have not come across this in medical terms. A commonly used medical term would be more appropriate.

Just curious why antibiotics were used. Any necrosis identified in CT scan?

Instead of primary disease of lipid metabolism consider changing to primary disorders

Line 136: usually the mortality rate is not this high in children. Consider rephrasing with other reference. 

Author Response

Thank you for your review. Regarding your questions, I want to mention the following aspects:

  • The patient experienced pain localized all over the abdomen, but the intensity was higher in the periumbilical region, without any radiating pain in the back.
  • Fasting lipid panel was collected after admission in the Pediatric Department. The values were: triglycerides (1359 mg/dL) and total cholesterol (538 mg mg/dL) with HDL- 35.6 mg/dL, LDL- 214.8 mg/dL and VLDL- 54 mg/dL.
  • In our lab, the normal values for pancreatic enzymes are 5-31 U/L for lipase and 22-50 U/L for amylase. The values mentioned in the text were obtained at admission. Within a week, his pancreatic enzymes levels were  as high as 153 U/L for lipase and 176 U/L for lipase. We will add the information in the manuscript as well.
  • Obesity is defined as a BMI at or above the 95th percentile for children and teens of the same age and sex. CDC does not define any classes of obesity in children.
  • The word "diffuse" means "widespread" and refers to pain that is localized all over the abdomen, or at least in many areas.
  • The use of antibiotics was prophylactic.
  • Thank you for your suggestion, we will make the change in our manuscript.
  • We will do more research regarding the mortality rate in children and make the necessary changes in our manuscript.

Reviewer 2 Report

The present study identifies a novel genetic variant of the LIPC gene, a gross deletion of the genomic region encompassing exon 1.

Major comments:

Paper is based on the relatively rich literature (37 items, but only 30% of used publications are from the last TEN years). Use more novel publications.

The aim of the study is missing in the paper.

Lack of LIMITATIONS of study (at the end of the paper).

Minor comments:

Instead of:

His maternal grandfather has type II diabetes.

Should be:

His maternal grandfather has type 2 diabetes mellitus.

Instead of:

The most striking abnormality was a marked elevation of triglycerides (1359 mg/dL) and total cholesterol (538 mg mg/dL) with HDL- 35.6 mg/dL, LDL- 214.8 mg/dL and VLDL 54 mg/dL.

Should be:

The most striking abnormality was a marked elevation of triglycerides (1359 mg/dL) and total cholesterol (538 mg/dL) with HDL- 35.6 mg/dL, LDL- 214.8 mg/dL and VLDL - 54 mg/dL.

Instead of:

Liver enzymes including ALT, AST, gamma-GT, ALP, and total bilirubin levels were within normal ranges.

Should be:

Liver enzymes including ALT, AST, gamma-GT, ALP, and total bilirubin levels were within referenced ranges.

Instead of:

  1. Conclusions This section is not mandatory but can be added to the manuscript if the discussion is unusually long or complex.

Should be:

This section is not mandatory but can be added to the manuscript if the discussion is unusually long or complex.

Author Response

Thank you for your review. Regarding your questions, I want to mention the following aspects:

  • Few papers are available in the literature considering the rarity of the disorder. However, we will try to find and use more novel publications on the subject.
  • Thank you, we will add the aim of the paper to the manuscript: to present the rare case of acute pancreatitis in a pediatric patient determined by an extremely rare cause of hypertriglyceridemia, a novel variant of the LIPC gene, reported in only a few families to date.
  • We will complete our manuscript with limitations of the study.
  • Thank you for the minor comments, we will consider and apply them in our manuscript.

Round 2

Reviewer 1 Report

Authors have addressed the queries appropriately. Just a pointer, diffuse abdominal pain is all over the abdomen. Localized if pain is in one or few of the 9 quadrants and if the pain is in all over the abdomen, this will be considered as diffuse abdominal pain. 

Although CDC does not classify severity of obesity in children, various organization including AAP recognizes these terms and they have implications on outcomes.